# Functional Load Capacity of Teeth with Reduced Periodontal Support: A Finite Element Analysis

**DOI:** 10.3390/bioengineering10111330

**Published:** 2023-11-18

**Authors:** Marco Dederichs, Paul Joedecke, Christian-Toralf Weber, Arndt Guentsch

**Affiliations:** 1Policlinic of Prosthetic Dentistry and Material Science, Centre for Dental Medicine, Jena University Hospital, D-07743 Jena, Germany; marco.dederichs@med.uni-jena.de; 2Department of Engineering and Industrial Design, Magdeburg-Stendal University of Applied Sciences, D-39114 Magdeburg, Germanychristian-toralf.weber@h2.de (C.-T.W.); 3School of Dentistry, Marquette University, Milwaukee, WI 53233, USA

**Keywords:** finite element analysis, periodontium, bone loss, attachment loss, maxilla, mandible, teeth

## Abstract

The purpose of this study was to investigate the functional load capacity of the periodontal ligament (PDL) in a full arch maxilla and mandible model using a numerical simulation. The goal was to determine the functional load pattern in multi- and single-rooted teeth with full and reduced periodontal support. CBCT data were used to create 3D models of a maxilla and mandible. The DICOM dataset was used to create a CAD model. For a precise description of the surfaces of each structure (enamel, dentin, cementum, pulp, PDL, gingiva, bone), each tooth was segmented separately, and the biomechanical characteristics were considered. Finite Element Analysis (FEA) software computed the biomechanical behavior of the stepwise increased force of 700 N in the cranial and 350 N in the ventral direction of the muscle approach of the masseter muscle. The periodontal attachment (cementum–PDL–bone contact) was subsequently reduced in 1 mm increments, and the simulation was repeated. Quantitative (pressure, tension, and deformation) and qualitative (color-coded images) data were recorded and descriptively analyzed. The teeth with the highest load capacities were the upper and lower molars (0.4–0.6 MPa), followed by the premolars (0.4–0.5 MPa) and canines (0.3–0.4 MPa) when vertically loaded. Qualitative data showed that the areas with the highest stress in the PDL were single-rooted teeth in the cervical and apical area and molars in the cervical and apical area in addition to the furcation roof. In both single- and multi-rooted teeth, the gradual reduction in bone levels caused an increase in the load on the remaining PDL. Cervical and apical areas, as well as the furcation roof, are the zones with the highest functional stress. The greater the bone loss, the higher the mechanical load on the residual periodontal supporting structures.

## 1. Introduction

Increasing the loss of periodontal attachment is associated with an increased loss of function, tooth mobility, and, eventually, tooth loss [1]. A strong correlation between attachment and alveolar bone loss and tooth mobility has been well-known for decades [2]. Recent data suggest that different types of teeth respond differently to attachment and bone loss regarding tooth mobility [3]. The loss of periodontal supporting structures results in the less stable mechanical support of the teeth. This has a negative effect on the masticatory performance and esthetic appearance due to anteriorly flared upper anterior teeth and quality of life for the patient [4,5]. For the practitioner, on the other hand, a reduced attachment results in more difficult conditions. For instance, an orthodontist must take into account that teeth with reduced attachment show an altered allocation of force transmission [6]. At the same time, less force is required for tooth movement in order not to damage the apical neuro-vascular bundle or the tooth–PDL–bone complex [7,8,9]. A periodontist may face increased difficulties in the application of regenerative materials, particularly in the furcation area [10]. From a prosthetic standpoint, periodontal-compromised teeth can pose a risk if they are included in the planning and implementation of dental restorations, especially if these compromised teeth are supposed to serve as abutment teeth [11].

Little is known about the threshold of when and how the loss of attachment affects biomechanical stability. The stress and strain created in the periodontal tissue in accordance with the morphological alteration of the structures must be considered if the interaction of reduced periodontal support with a mechanical function is investigated [12]. This is highly challenging because the periodontal complex, consisting of the cementum, periodontal ligament (PDL), and adjacent alveolar bone, is a unique tissue complex. Specific material properties have to be considered since the PDL is a visco-elastic material that allows for the distribution and absorbtion of the pressure produced during masticatory functions and other forms of contact between teeth, which act as tensile forces along the alveolar bone and can have a protective and restorative effect on the surrounding bony structures [13].

A method that allows us to analyze stress distribution in soft and hard tissues is the finite element (FE) method [14]. Commonly used as an engineering tool to evaluate mechanical properties in automotive and aircraft engineering, the FE method quickly has relevant applications in medicine and dentistry as well. The FE method was applied to analyze the stress in periodontal tissues during tooth movement [15,16], to investigate stress and strain in the bone surrounding dental implants [17], to evaluate bone remodeling effects during orthodontic treatments [18], to assess the influence of different dental crown materials on the biomechanical properties of the tooth structure [19], and preliminary research was performed to investigate the change in stress distribution under a simulated bite force in a periodontally compromised tooth [12]. A wide range of possible applications confirms that the FE method, which has been implemented and adequately described for decades, is relevant for answering current questions about biomechanical properties. The great advantage of the FE method is that it can completely and non-invasively provide relevant data about the force distribution in the physiological as well as in the pathological system.

In particular, the simulation of loads in maximum ranges and their effect on the investigated structures makes the FE method an interesting and relevant tool in the long-term assessment of different dental situations, from which relevant estimations on the survival rate of teeth subjected to parafunctional loading can be drawn [20]. This makes the FE method particularly interesting for use in dentistry since a large number of different mechanical parameters and morphological properties interact in a very small space.

Studies that simulate the mechanical behavior of oral structures require a highly complex analysis because of the differences in the material properties of the different tissues in the oral system. Such studies are of fundamental importance for acquiring knowledge on how these elements function together under physiological or pathological conditions to optimize prognosis and treatment outcomes [21,22].

The purposes of this study are (a) to determine the functional load capacity of single- and multi-rooted teeth with different levels of periodontal support using a finite element model and (b) to investigate three-dimensional stress distribution in the periodontal tissues under a masticatory load. We hypothesize that single and multi-rooted teeth distribute their load and strain differently and respond dissimilarly to periodontal attachment loss.

## 2. Materials and Methods

Cone-beam computer tomographic (CBCT) data were used to create 3D models of individual dental structures. The CBCT was taken solely due to a medical indication and not for research purposes. The volunteer made his CBCT available for teaching or research purposes. The local IRB committee (Office of Research Compliance, Human Subjects and Radiation Safety) authorized the use of the CBCT (DT-041).

The DICOM tags of the header file were edited to anonymize the patient and to fit the resolution of each grayscale image for use in additional software (Figure 1).

Segmentation was performed using InVesalius 3.0 (an open-source medical framework) for larger areas and ITK-SNAP 3.8.0 (open-source interactive segmentation software) to generate each individual tooth. A threshold filter was applied to separate the marrow and cortical bone in InVesalius 3.0. The semi-automatic procedures in ITK-SNAP were used to determine the area of interest and to apply the implemented fast-growing algorithm. After processing all structures, Geomagic Wrap 2015 (3D Systems, Moerfelden-Walldorf, Germany) was used to adapt the quality of the generated surface mesh of each structure, including the mesh doctor, removing artifacts, filtering redundancies, and converting the surface description into the CAD file format. Furthermore, the number of triangles for use was specified, and the PDL layer was created. To perform this, the outer layer of the dentin was thickened by 0.1 mm and cut out as a new volumetric model [23]. A layer thickness between dentin and PDL was defined as cementum in three steps from 50 µm at the CEJ and 150 µm at the apex [24]. To investigate the behavior of a reduced PDL layer, 10 additional models of single- and multi-rooted teeth were created (Figure 2), and each model was given a reduction of 1 mm in its PDL layer height along the root axis.

The static structural FE analysis was performed using Ansys Workbench 17.2 (ANSYS, Canonsburg, PA, USA). The number of used tetrahedral elements was 1,508,356 and 847,242 nodes. The contact surfaces between the individual structures were set as bonded except for the occlusal contact areas, where it was set as rough and adapted to touch. This approach ensures the presence of tensile and pressure stress in the PDL layer and additionally allows for tooth movement. The support was set to be fixed on the upper side of the maxilla, and a cylindrical support was used for the mandible. Figure 3 depicts a CAD-formatted 3D model with a visual representation of the direction of the force application. The force was applied stepwise, simulating the muscle approach of the masseter muscle in 70 analysis steps from 0 N up to 700 N in the cranial and 350 N in the ventral direction. As a result, a maximum force of 782.6 N was applied compared to that of van Eijden, 1991 [25].

The occlusal contact area is illustrated in Figure 4. Sufficient and evenly distributed occlusal contacts were chosen since it was suggested that the amount of the occlusal contact areas is critical for masticatory performance [26].

The occlusal contact areas are influenced by the occlusal surface structure. Maxillary contact areas are illustrated in red, mandibular contact areas are illustrated in blue. For technical reasons, tooth 15 (upper left second molar) appears in yellow color as a visual indicator of the selected tooth within the CAD software (Geomagic Wrap 2015, 3D Systems, Moerfelden-Walldorf, Germany).

For mesh generation, the transition was set to be slow, and a manual mesh size was added to suffice the convergence test. To compare the results of each PDL layer, the pressure distribution between the cortical bone and the PDL layer was used by implementing the contact tool. Further quantitative (pressure, tension, and deformation) and qualitative (color-coded images) data were recorded and analyzed. The material properties of various tissues were assigned according to Table 1.

## 3. Results

A qualitative analysis was performed to illustrate, in color-coded images, the stress distribution areas in accordance with the impact of the influencing force. Furthermore, a quantitative analysis was performed to visualize the amounts of force applied.

### 3.1. Three-Dimensional Stress Distribution in the Periodontal Tissues

Teeth with full periodontal support demonstrated high stress in the cervical and apical regions under functional loading. This was more prominent in canines, premolars, and molars than in incisors (Figure 5).

The teeth with the highest load capacities were the upper and lower molars (0.4–0.6 MPa), followed by the premolars (0.4–0.5 MPa) and canines (0.3–0.4 MPa) when vertically and occlusally loaded. While the described pressure distribution in the canine, premolar, and molar regions was almost equally distributed over the cervical and apical regions, the furcation regions of the molars, in particular, showed punctual load peaks of up to 0.7 MPa. The incisors, on the other hand, showed a uniform pressure distribution over the entire PDL surface without regional stress peaks.

The distribution of the forces under an axial load (simulated bilateral contraction of the masseter muscle) identified the upper first molars and the lower first and second molars with the highest loading, followed by premolars and canines, with insignificant forces calculated for incisors (Figure 6).

### 3.2. Functional Load Capacity of Single- and Multi-Rooted Teeth with Different Levels of Periodontal Support

The furcation area was identified as a stress zone in multi-rooted teeth (Figure 5), as seen in the stress distribution of alveolar housing (Figure 7).

A stepwise reduction in the PDL support resulted in an increase in the load in the remaining PDL area with an increase especially in the apical area. Since the furcation area in molars is no longer available for load distribution, the remaining PDL structure was loaded to a greater extent, which revealed an increased pressure load, particularly at the apex (Figure 8).

When the periodontal support was reduced (in 1 mm steps), the remaining PDL showed higher pressure values. The maximal load achieved for a single-rooted tooth with full periodontal support structures was 0.48 MPa. With a stepwise reduction, this load increased in a logarithmic-like manner, with 0.52 MPa for a 2 mm reduction, 0.59 MPa for a 4 mm, and 0.77 MPa for a 6 mm. At a 7 mm reduction, the load was determined with 0.90 MPa, after which the FEM predicted the failure of the tooth. The multi-rooted tooth followed a similar pattern with 0.40 MPa when full periodontal support was simulated at 0.42 MPa for a 2 mm reduction, 0.50 MPa for a 4 mm, 0.60 MPa for a 6 mm, and 0.85 MPa for an 8 mm reduction. When the PDL was reduced to more than 8 mm, the model predicted the failure of the tooth (Figure 9).

## 4. Discussion

The complex science of how forces are distributed during mastication is of fundamental interest in dentistry [33]. The present study impressively demonstrates the periodontal force distribution within the entire dentition during axial force transmission, as it occurs during tooth contact due to the contraction of the masseter muscles. Whereas recent studies have focused on individual teeth, such as incisors [5,34,35,36], canines [37], premolars [38,39], or molars [40], the present study was able to consider the entire periodontium and, thus, describe the different effects on single-rooted and multi-rooted teeth at the same time. This holistic approach confirmed the hypothesis that single and multi-rooted teeth distribute their load and strain differently and respond dissimilarly to periodontal attachment loss.

In periodontal health, the main force distribution is in the region of the molars, followed by the premolars and the canines. This finding is consistent with observations by van Eijden and is due to the local efficiency of the masseter muscles involved in jaw closure [25]. An anteriorly decreasing power arm due to the anatomical muscle course is reflected in a similarly reduced power transmission anteriorly. The investigations of occlusion formation demonstrated that incisors may play a guiding role via the proprioceptors of the periodontium as components of the neuromuscular control of mandibular movement [33]. However, with a rising level of force, the masticatory force distribution shifts toward the molar area, with a decreased relative portion of forces in premolar and anterior regions [33,41]. The masticatory center, where the highest masticatory force is applied during habitual occlusion, is in the molar region, and the lowest force is seen in the anterior region [42]. Remarkably, in the present FE analysis, the furcation areas of the molars were found to play a significant role in force absorption. While the highest stress areas in the molars, premolars, and canines under axial loading were in the cervical and apical regions, the furcation areas of the molars also exhibited significantly increased stress areas. Thus, in their role as the main masticatory center in physiologically healthy dentition, the furcations of the maxillary and mandibular first molars appeared to play a significant role in force transmission.

A gradual reduction in the bony attachment and, thus, also in the available PDL showed an approximately logarithmic increase in the transmitted force as well as a shift in the stress ranges. While the force was mainly transmitted cervically and apically in the periodontally healthy dentition, a shift to the apical region took place after a reduction in the attachment. These findings are in line with the finite element analysis conducted by Reddy and Vandana, who investigated the force distribution on loaded incisors [43]. This effect is particularly evident in the molar region as soon as the bony support of the furcation area is lost.

When considering the changed force transmission to the reduced attachments, a critical problem arose. A horizontal reduction in the periodontium resulted in larger maximum values for tensile and compressive forces. Reduced periodontal support led to an increase in stress in the remaining structures [44]. Compressive forces generally lead to remodeling processes within the surrounding bone structures and cause bone resorption at this site. Tensile stresses, on the other hand, stimulate the bone to grow [45,46]. The forces applied during mastication are controlled via mechanoreceptors in the periodontium under physiological conditions [47,48]. If the periodontal function is impaired—as in the case of reduced support structures—the function of the mechanoreceptors is disturbed. In addition, the stress on the fibers present in the periodontium increases [49], which results in the normo-function being exceeded and physiological remodeling processes no longer functioning. It is reported that a critical stress threshold is reached after 30% of periodontal bone is lost [38]. However, the traumatic loading of the PDL does not appear to directly cause the degradation of the surrounding bone, as investigations by Ona and Wakabayashi showed . Rather, the risk for progressive bone resorption seems to lie in the constant repetition of pressure on the surrounding apical bone [50]. In addition, a long-term study on factors leading to tooth loss due to periodontal disease highlighted that in addition to the presence of severe periodontal disease, a combination of smoking and the presence of bruxism could significantly increase the risk of tooth loss [51]. The presence of bruxism alone could already result in a three-fold increase in vertical bone resorption [52], highlighting the influence of increased sustained pressure on bone resorption. This not only results in increasing discomfort and reduced quality of life for the patient but significantly reduces periodontal attachment, which can be a challenge for practitioners as well. Furcation-involved molars can be resistant to treatment and can lead to a poor prognosis of the tooth [10]. In particular, molars with advanced furcation involvement are considered to be at particular risk of loss [53]. From a prosthetic point of view, the inclusion of periodontally compromised teeth also represents risks. Fixed dentures can lead to the additional overloading of the remaining PDL and significantly jeopardize the long-term prognosis of the planned denture [54].

Even though the FEA is the only way to show stress distribution in the different tissues of the periodontium and, therefore, the present study reveals a comprehensive view of stress distribution on the PDL of the upper and lower dentition under the consideration of different levels of attachment loss, there are limitations. Since finite element analyses are based on mathematical models, they must be considered as simulations of a biological system. Individual parameters may differ from individual to individual and may influence the results. Therefore, it is difficult to compare results from FE analyses, as the results are highly dependent on the input of the tissue-related physical parameters [35]. However, a precise simulation of the PDL represents a major challenge [20]. Due to the complex visco-elastic properties, this structure, as a non-linear system, is highly error-prone in the calculation due to its poor convergence behavior. Additionally, other material-related aspects must be considered. The interaction of material factors and the geometry and boundary conditions promotes the risk of non-linear behavior. Accordingly, it must be taken into account that non-linearity due to geometry changes cannot be excluded in a complex tooth structure. In the present study, all material values were assumed to be linearly elastic. Therefore, it can be assumed that the response of this system is also linear. To reduce this risk and generate a converging system with less error potential, 70 load steps were chosen for the force application. Since there is little consensus regarding the simulation of the PDL as a visco-elastic system, the approach followed in the present study can be considered as a proof of concept on which subsequent investigations can build and implement non-linear material behavior. Although there may be differences in the reported total values of different studies, the trend of these results gives a clear indication of the consequences of attachment loss. Further analysis should be conducted to support the results, especially with the developments in machine learning and its applications in treatment planning.

## 5. Conclusions

Functional stress was highest under an axial masticatory load in the molars, followed by premolars and canines. In the case of full periodontal support, the zones of highest stress were in the cervical and apical areas, whereas, in the multi-rooted teeth, a significantly increased pressure load was also found in the furcation area. When the area of the PDL was successively reduced to simulate attachment loss, the pressure load on the remaining PDL increased approximately logarithmically and shifted significantly toward the apical region. A single-rooted tooth with an attachment loss of >7 mm was predicted to fail, and a multi-rooted tooth with >8 mm attachment loss was predicted to fail. These findings suggest that there are thresholds of 7 mm in single-rooted teeth and 8 mm in multi-rooted teeth that can determine when a tooth is hopeless due to periodontal attachment loss, which can help the clinician in decision making when to extract. These thresholds can also be applied to assess the quality of a tooth as a potential abutment tooth for prosthetics.

## Figures and Tables

**Figure 1 bioengineering-10-01330-f001:**
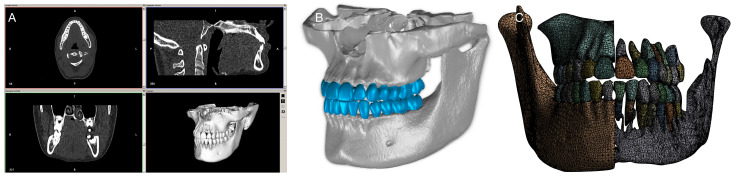
From the DICOM format to Finite Element Mesh. CBCT data (**A**) were transformed into the CAD file format as a segmented 3D model of the dento–alveolar complex (**B**) followed by the creation of surface mesh (**C**).

**Figure 2 bioengineering-10-01330-f002:**
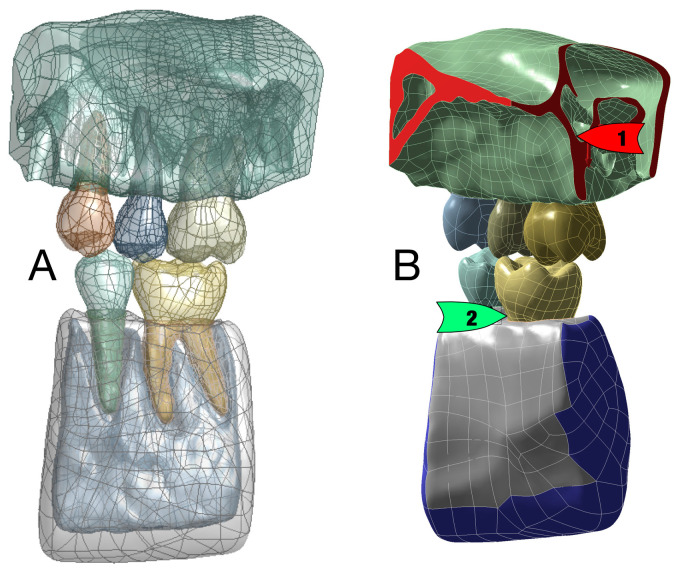
CAD-formatted 3D model (**A**) including posterior single- and multi-rooted teeth for = reduced periodontium analysis. The FEA model (**B**) shows that the forces were applied from the maxillary teeth and the maxilla (1, highlighted in red) toward the teeth and bone (blue) in the mandible (2).

**Figure 3 bioengineering-10-01330-f003:**
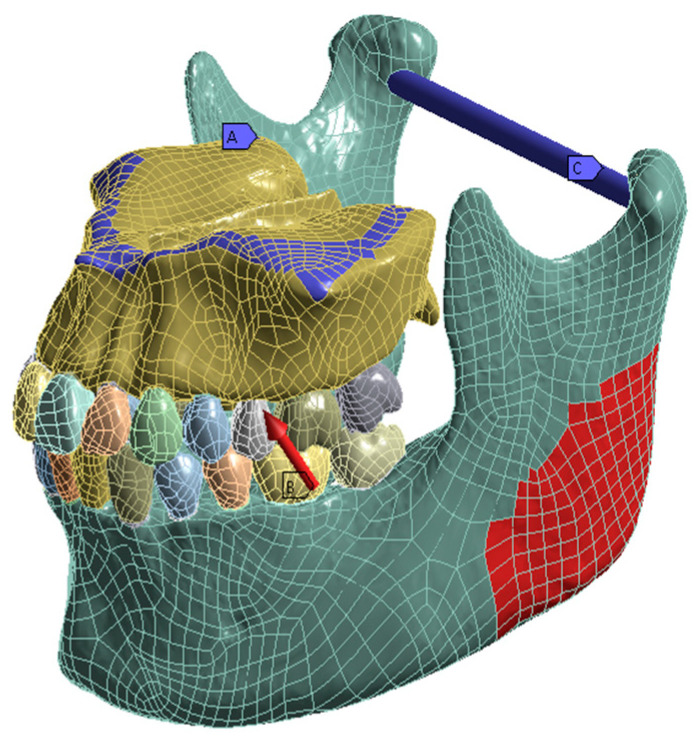
CAD-formatted 3D model with indications of loading and boundary conditions. The maxilla (A) was set as fixated. The red arrow at the lower left first molar illustrates the direction in which the force was applied with maximal 700 N in the cranial and 350 N in the ventral direction (B) towards the maxilla. The blue axis (C) connecting the two condyles represents the axis of rotation. The red area reflects the insertion of the masseter muscle.

**Figure 4 bioengineering-10-01330-f004:**
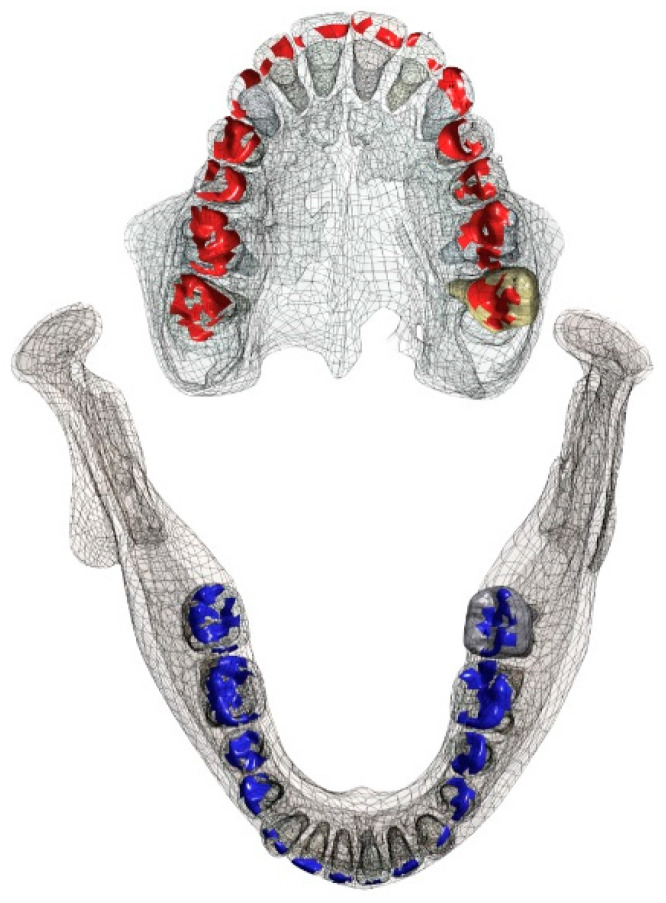
Areas of initial occlusal contact. In red the contacts on the maxillary teeth and in blue the contact area of the teeth in the mandible.

**Figure 5 bioengineering-10-01330-f005:**
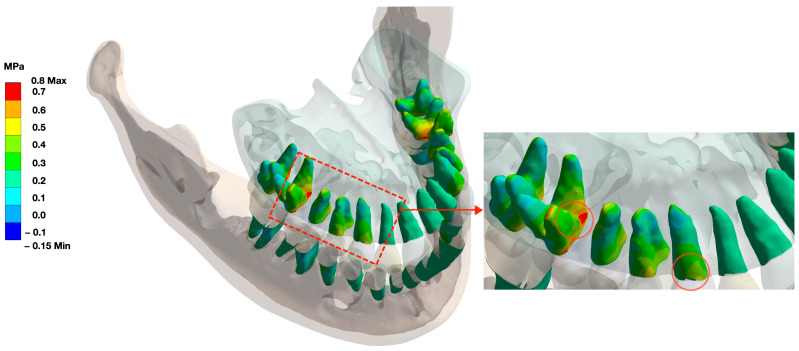
Pressure distribution under functional loading on teeth with full periodontal support. Areas of stress are colored yellow and red. The pressure distribution appears to be most prominent in the cervical area of the molars, premolars, and canine teeth (circle in the magnified area). In molars, the furcation roof (circled in the magnification) shows the most stressed area (see also Figure 6).

**Figure 6 bioengineering-10-01330-f006:**
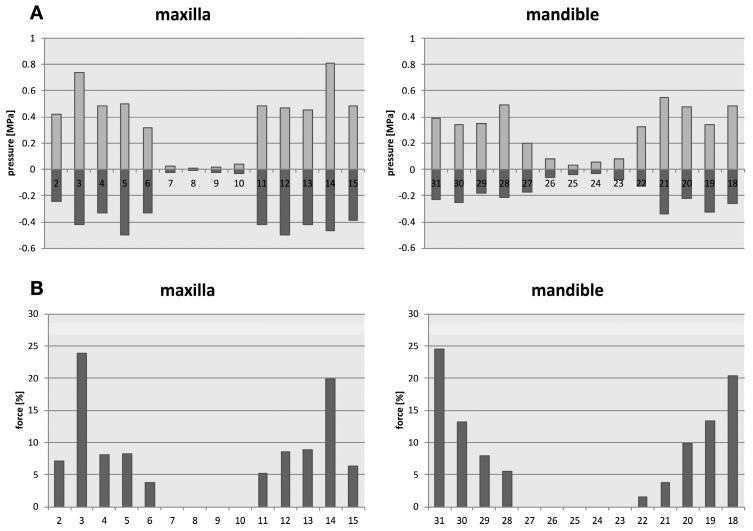
Distribution of forces under an axial load (Pressure stress—positive values, tensile stress—negative values). Under a simulated axial load, the pressure in the periodontium of the individual teeth is shown in the upper row diagrams (**A**). The upper first molars, and the second and first lower molars thereby carry the most force, followed by the premolars, and the canines (**B**). The anteriors are only minimally impacted by axial load. Teeth are numbered according to the Universal Numbering System.

**Figure 7 bioengineering-10-01330-f007:**
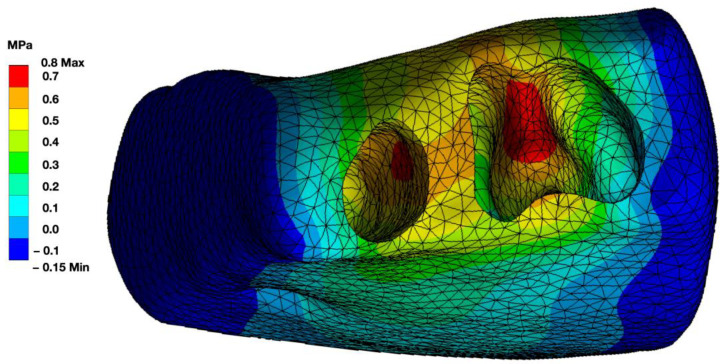
Highlighted illustration of the bony housing of a single-rooted and a multi-rooted tooth during functional load. The area of the interradicular bone appears to be a highly stressed region.

**Figure 8 bioengineering-10-01330-f008:**
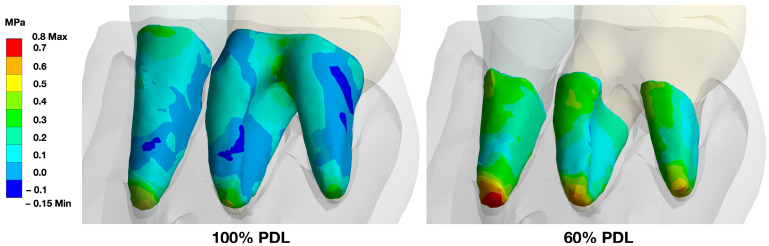
Illustration of changes in force distribution when reducing the periodontal support. In fully periodontally supported teeth, the highest stress under loading conditions was found at the cervical area, the apical area, and in molars in the furcation area. When the periodontal support was reduced (shown here for 60% attachment), the remaining periodontally supported root structure carried the load, and the stress zones were found to be more evenly distributed along the root, with a highlight at the apices of the teeth.

**Figure 9 bioengineering-10-01330-f009:**
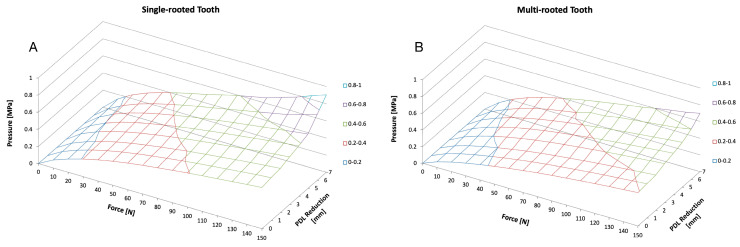
Graphical illustration of the correlations between increasing force and a reduction in PDL on the pressure development of a single (**A**) and a multi-rooted (**B**) tooth (as shown in Figure 7). The illustration depicts a single-rooted tooth (left hand) and a multi-rooted tooth (right hand) as shown in Figure 8. In general, pressure on PDL increases with an increasing force as well as with an increasing PDL reduction.

**Table 1 bioengineering-10-01330-t001:** Material characteristics of tissues used for finite element analysis.

Material/Component	Elastic Modulus (MPa)	Poisson Ratio
Enamel [27]	84,000	0.33
Dentin [28]	18,600	0.31
Pulp [29]	2.07	0.45
PDL [30]	68.9	0.45
Gingiva [31]	3.0	0.45
Cortical bone [32]	13,700	0.3
Cancellous bone [32]	1370	0.3

## Data Availability

The data are available upon reasonable request.

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
