# Peer review of "Functional Load Capacity of Teeth with Reduced Periodontal Support: A Finite Element Analysis"

_bioengineering, 2023, doi:10.3390/bioengineering10111330_

Round 1

Reviewer 1 Report

Comments and Suggestions for Authors

Dear Authors,

I read with interest your work "Functional Load Capacity of Teeth with Reduced Periodontal Support: A Finite Element Analysis”.

In particular, the group aimed to study the functional load capacity of single and multi-rooted teeth with different levels of periodontal support using a finite element model and to investigate the three-dimensional stress distribution in the periodontal tissues in masticatory load. They hypothesize that single and multi-rooted teeth distribute load and strain differently and will respond dissimilar to periodontal loss.

While the results and conclusions are not novel, they are based on solid statistical data and help to extend the knowledge on the topic, which may be beneficial in the future.

Therefore, I have no further comments against the manuscript.

Author Response

Manuscript ID bioengineering-2638967

Functional Load Capacity of Teeth with Reduced Periodontal Support: A Finite Element Analysis

Dear Professor Dr. Kwong Ming Tse,

We would like to thank Bioengineering for the careful evaluation of our manuscript, and we thank the reviewers for their valuable suggestions. The manuscript has been revised according to reviewers’ comments and the paper has much improved because of the changes that were suggested.

Comments and Suggestions from Reviewer 1:

Dear Authors,

I read with interest your work "Functional Load Capacity of Teeth with Reduced Periodontal Support: A Finite Element Analysis”.

In particular, the group aimed to study the functional load capacity of single and multi-rooted teeth with different levels of periodontal support using a finite element model and to investigate the three-dimensional stress distribution in the periodontal tissues in masticatory load. They hypothesize that single and multi-rooted teeth distribute load and strain differently and will respond dissimilar to periodontal loss.

While the results and conclusions are not novel, they are based on solid statistical data and help to extend the knowledge on the topic, which may be beneficial in the future.

Therefore, I have no further comments against the manuscript.

Response: We would like to thank the reviewer very much for the time and effort in reviewing and evaluating our manuscript. We are very pleased with the reviewer’s positive assessment and that our study met the expectation of the reviewer.

Text Change: none

Reviewer 2 Report

Comments and Suggestions for Authors

This study aims to explore the impact of the loss of PDLs, connecting teeth and the alveolar bone, on the distribution and transmission of mechanical loads in mastication. This topic and results hold significant clinical relevance and attract considerable attention. A well-designed approach has been adopted, melding CBCT scans with finite element modelling to investigate the issue in a systematic approach with controlled variables. The results were presented in response surface forms, highlighting the significance of PDL contacts when compared to the load magnitude. Nonetheless, before delving into the results, a few points warrant clarification.

Abstract:

Line 17: How was the cementum segmented from the dentin and enamel? It doesn’t appear to have its own geometry or property in the body text.

Method:

Line 51-52: what is ‘pressure force’? Separately, these are two very different physical quantities. What is the role of the viscous component of PDL properties here? Please elaborate.

Line 106: Please include schematics to visualise the loading and boundary conditions assigned to the model. What is the justification for applying such a magnitude of force to a clinical scenario with periodontitis? Is there any later literature, with improved techniques, about measuring occlusal loads or muscular forces suggesting different magnitudes? Was the force applied with uniform steps or adjusted to allow convergence?  

Figure 2: Can you explain why the contact surfaces look different between maxillary and mandibular dentitions? What is the yellow colour on the maxillary molar?

Table 1: Introduction mentions the viscoelastic property of PDL. Why is a linear elastic material model adopted for one of the most critical components in the complex?

What is ‘well-distributed contact’? How was the contact condition assigned? Frictionless or what? What is the justification?

Is there any information about how the tooth-PDL-bone contact was altered? Uniformly or following the contours of PDLs?

Results:

Figure 3: The plot seems to be a stress contour (MPa in unit) at the roots – why is it called ‘force distribution’ (N in unit)? What type of stress is that? What does the negative stress mean? Separate and zoom-in views showing stress extremities would be appreciated.

Same comments to Figure 4.

Section 3.1 is already referring to the numerical outcomes in the figures to draw the observation. It is suggested to re-organise the result sections based on the focuses, e.g. global load distribution, single-root vs. multiple-root,  effects of PDL contact/mastication load magnitude, etc.

What is the definition of ‘the highest load capacities’? Are the observations referred to Figure 6a/b? These results need to be addressed in more detail. What do the positive/negative pressures mean here? If they are max/min values of the pressures developed under a given mastication scenario, how would these values infer the capacities of loading?  

Line 173: which teeth are reported and presented in Figure 7?

Comments on the Quality of English Language

Some improvements must be made to the clarity of the language and the technical details. 

Author Response

Manuscript ID bioengineering-2638967

Functional Load Capacity of Teeth with Reduced Periodontal Support: A Finite Element Analysis

Dear Professor Dr. Kwong Ming Tse,

We would like to thank Bioengineering for the careful evaluation of our manuscript, and we thank the reviewers for their valuable suggestions. The manuscript has been revised according to reviewers’ comments and the paper has much improved because of the changes that were suggested.

*Please note that changes have been highlighted in red in the revised manuscript.

Comments and Suggestions from Reviewer 2:

This study aims to explore the impact of the loss of PDLs, connecting teeth and the alveolar bone, on the distribution and transmission of mechanical loads in mastication. This topic and results hold significant clinical relevance and attract considerable attention. A well-designed approach has been adopted, melding CBCT scans with finite element modelling to investigate the issue in a systematic approach with controlled variables. The results were presented in response surface forms, highlighting the significance of PDL contacts when compared to the load magnitude. Nonetheless, before delving into the results, a few points warrant clarification.

1) Abstract:

1.1) Line 17: How was the cementum segmented from the dentin and enamel? It doesn’t appear to have its own geometry or property in the body text.

Response: We thank the reviewer for the time and effort taken in reviewing our manuscript. We hope that we have implemented all the suggestions and indications to the full satisfaction of the reviewer.
Regarding the geometry of the cementum, due to the CBCT voxel sizes of 150 µm the root cementum could not be segmented from the DICOM data in the present study. A layer thickness between dentin and PDL was defined as cementum in three steps from 50µm at the CEJ and 150µm at the apex (Jang et al. J Mech Behav Biomed Mater 2024; 39: 184-196). The material values (Young's modulus and transverse contraction coefficient) correspond to the values of the PDL.

Text Change: A respective note was added to the material and methodology section.

Materials and Methods: Further the number of triangles to be used was specified and the PDL layer was created. To do so, the outer layer of the dentin was thickened by 0.1mm and cut out as new volumetric model [14]. A layer thickness between dentin and PDL was defined as cementum in three steps from 50µm at the CEJ and 150µm at the apex (Jang et al. J Mech Behav Biomed Mater 2024; 39: 184-196). To investigate the behavior of a reduced PDL layer 10 additional models of single- and multi-rooted teeth were created of which each model has a reduction of 1mm in PDL layer height.

2) Method:

2.1) Line 51-52: what is ‘pressure force’? Separately, these are two very different physical quantities. What is the role of the viscous component of PDL properties here? Please elaborate.

Response: Thank you very much for this comment. We admit that the term 'pressure force' is an unfortunate or incorrect translation. What was meant was "pressure distribution". Pressure is force per area. Therefore, the terms pressure and force are directly related but incorrect presented in the manuscript. The aim is to achieve an optimum pressure distribution, which also means a uniform force distribution if the surface area remains constant. In terms of the role of the viscous component of PDL properties: The mention of viscoelastic properties is intended to highlight the complex structure of PDL. However, linear elastic material properties are assumed for the calculation. This simplification was accepted in order to have fewer variables and thus sources of error in the calculation.

Text Change: The entire manuscript was searched for the incorrect term ‘pressure force’ and corrected to its actual meaning, ‘pressure distribution’.

Introduction (lines 55-59): Specific material properties have to be considered, since the PDL is a visco-elastic material which allows for the distribution and absorbtion of pressureforces produced during masticatory functions and other tooth contacts into tensile forces along the alveolar bone which can have a protective and restorative effect on the surrounding bony structures [13].

Figure 6 (lines 198-203): Distribution of the forces under axial load (Pressure stress – positive values, tensile stress – negative values). Under simulated axial load, the pressure forces in the periodontium of the individual teeth are shown in the upper row diagrams (A). The upper first molars, and the second and first lower molars therby carry the most forces, followed by the premolars, and the canines (B). The anteriors are only minimally impacted by axial load. Teeth are numbered according to the Universal Numbering System.

2.2) Line 106: Please include schematics to visualise the loading and boundary conditions assigned to the model. What is the justification for applying such a magnitude of force to a clinical scenario with periodontitis? Is there any later literature, with improved techniques, about measuring occlusal loads or muscular forces suggesting different magnitudes? Was the force applied with uniform steps or adjusted to allow convergence?

Response: Thank you for this relevant suggestion. The magnitude and direction of the simulated force were taken from the publication by van Eijden [15]. To the best of our knowledge, there is no recent study that could be applicable for our investigation. The maximum values that were described by van Eijden were set as the maximum vectorial force to be calculated. A central question was how this load affects different periodontal conditions in detail, since to the best of our knowledge this question had never been investigated before to a comparable extent.

Regarding the question of whether the force was applied in uniform steps or adjusted: the step size was chosen to be uniform. Convergence was achieved by choosing a small step size of 70 analysis steps. It was considered that in the case of a tooth structure, nonlinearity due to changes in geometry (changed contact) cannot be excluded. The determination of 70 analysis steps thus enabled a converging system that remains calculable.

Text Change: Materials and methods (lines 132-134): The support was set to be fixed on the upper side of the maxilla and a cylindrical support was used for the mandible. Figure 3 depicts a CAD-formatted 3D model with a visual representation of the direction of force application.

New Figure 3 (lines 138-145): An illustration to visualize the loading and boundary conditions assigned to the model was added to the manuscript.

2.3) Figure 2: Can you explain why the contact surfaces look different between maxillary and mandibular dentitions? What is the yellow colour on the maxillary molar?

Response: The occlusal surfaces have different shapes/structures due to the meshing. The size of the individual surfaces is approximately the same. To select the surface pairings (occlusal surfaces) between the upper and lower jaw, the model was aligned in such a way that an open contact is created between the tooth pairings. The distance between them is not equal. This is due to the suspension of the mandible. To compensate for this effect, a pinball area has been defined. This mechanism can provide additional control over the geometric extent of various objects such as joints, remote points, springs, point masses, remote force, and remote displacement. The yellow color on the maxillary molar is a CAD feature that unfortunately cannot be disabled. In Ansys Workbench there must always be a selected activated object. In the present case this was the second molar in the left site.

Text Change: Figure 2 4: Areas of initial occlusal contact. The occlusal contact areas are influenced by the occlusal surface structure. Maxillary contact areas are illustrated in red, mandibular contact areas are illustrated in blue. For technical reasons, tooth 15 (upper left second molar) appears in yellow color as a visual indicator of the selected tooth within the CAD software.

2.4) Table 1: Introduction mentions the viscoelastic property of PDL. Why is a linear elastic material model adopted for one of the most critical components in the complex?

Response: Thank you, we appreciate this relevant question. The replication of the complex visco-elastic (strongly non-linear) material behavior is possible and can be found in the literature. However, from our point of view, there is a lack of relevant sources that shows a common consensus related to the replication of the PDL. Nonlinear systems always mean significantly higher computation times and are more error-prone due to the poor convergence behavior. This can have an even more serious effect with large data sets as used in the present study. The approach followed in the present study represents a proof of concept. Subsequent investigations can build on this and implement a nonlinear material behavior. Please see also our response to question #2.

Text Change: With additional consideration of the comments and recommendations from points 2.1 and 2.2, we have discussed the problem of comparing linear and nonlinear systems in more detail in the discussion section.

Discussion (lines 314-328): Therefore, it is difficult to compare results from FE analyses, as the results are highly dependent on the input of the tissue related physical parameters [35]. However, a precise simulation of the PDL represents a major challenge [20]. Due to the complex visco-elastic properties, this structure, as a non-linear system, is highly error-prone in the calculation due to its poor convergence behavior. Additionally, other material-related aspects must be considered. The interaction of the material factors, geometry and boundary conditions promotes the risk of nonlinear behavior. Accordingly, it must be taken into account that nonlinearity due to geometry changes cannot be excluded in a complex tooth structure. In the present study, all material values were assumed to be linearly elastic. Therefore, it can be assumed that the response of the system is also linear. To reduce this risk and generate a converging system with less error potential, 70 load steps were chosen for the force application. Since there is little consensus regarding the simulation of the PDL as a visco-elastic system, the approach followed in the present study can be considered as a proof of concept on which subsequent investigations can build and implement a nonlinear material behavior.

2.5) What is ‘well-distributed contact’? How was the contact condition assigned? Frictionless or what? What is the justification?

Response: We apologize for the imprecise description and thank you for pointing this out. What we were trying to refer to is a sufficient and even distribution of the selected occlusal surfaces. This selection is made manually by specifying the pinball area (see also comment 2.3). As contact condition was “frictional” used. From a mechanical standpoint, this prevents rigid body movement. In this application, the contact selection ensures that the occlusal surfaces can lift off from each other, but displacement is made more difficult. In a smooth contact condition, the tooth surfaces would "slide" too much and part of the applied force would be absorbed by the bearing on the condyles, which would be an unrealistic scenario. In addition, the contact selection counteracts divergence which allows for better computation. The choice of the optimum contact setting was determined by our own in-silico investigations. Virtual test setups were compared with in vitro investigations from chewing simulators.

Text Change: The imprecise wording has been removed and clarified.

Materials and Methods (lines 146-148): The occlusal contact area is illustrated in figure 2 4. Well-distributedSufficient and evenly distributed occlusal contacts were chosen since it is suggested that the amount of occlusal contact areas is critical for masticatory performance [26].

2.6) Is there any information about how the tooth-PDL-bone contact was altered? Uniformly or following the contours of PDLs?

Response: The reduction of the PDL was calculated in 1 mm steps by reducing the height of the PDL along the root morphology, thus following the contour of the root dentin. We have specified this in the Materials and Methods section.

Text Change: Material and Methods (lines 115-117): To investigate the behavior of a reduced PDL layer, 10 additional models of single- and multi-rooted teeth were created (Figure 2) of which each model was given a reduction of 1mm in PDL layer height along the root axis.

3) Results:

3.1) Figure 3: The plot seems to be a stress contour (MPa in unit) at the roots – why is it called ‘force distribution’ (N in unit)? What type of stress is that? What does the negative stress mean? Separate and zoom-in views showing stress extremities would be appreciated. Same comments to Figure 4.

Response: Thank you for this note. We agree that the term "force distribution" is incorrect here, as it is a ‘pressure load’ in the sense of normal pressure. We have corrected this. For a clearer presentation, we have added a second projection plane in the figure.

Text Change: Figure 3 5 (lines 178-182): Force Pressure distribution under functional loading on teeth with full periodontal support. Areas of stress are colored in yellow and red. The forcepressure distribution appears to be most prominent in the cervical area of the molars, premolars, and canine teeth (circle in the magnified area). In molars, the furcation roof (circled in the magnification) shows to be the most stressed area (see also Fig. 5).

3.2) Section 3.1 is already referring to the numerical outcomes in the figures to draw the observation. It is suggested to re-organise the result sections based on the focuses, e.g. global load distribution, single-root vs. multiple-root, effects of PDL contact/mastication load magnitude, etc.

Response: We recognize the problem that the complexity of the data presented the results section may be perceived as confusing or poorly structured. While a large part of the literature is oriented to individual teeth or tooth groups and deals with these in the FE analyses, the present study considered a much larger field. In order not to dilute the significance of the results by looking at them in too much detail, we have deliberately decided to focus on and discuss the most relevant results. This includes, on the one hand, the global consideration of single-rooted and multi-rooted teeth and, on the other hand, the special significance of the furcation areas of multi-rooted teeth. To address the reviewer’s comment, we re-organized the results to have first presented the maxilla/mandible model and the focus in a second segment on the single- and multi-rooted teeth.

Text Change: Reorganizing the result section.

3.3) What is the definition of ‘the highest load capacities’? Are the observations referred to Figure 6a/b? These results need to be addressed in more detail. What do the positive/negative pressures mean here? If they are max/min values of the pressures developed under a given mastication scenario, how would these values infer the capacities of loading?

Response: Thank you. We happily clarify. The highest load capacities are referred to the teeth as presented in Figure 6. The positive values represent pressure on the PDL and the negative values tension. We only analyzed the upwards movement of the mandible towards the maxilla as directed by the Masseter muscle to simulate axial load. We do not claim to have analyzed different mastication scenarios but can assume that shear forces in buccal-lingual and mesial-direction would even more increase the load and with that the stress on the periodontium. Especially a lateral movement would include teeth that are involved in canine or premolar guidance. Also, the load of the incisors will change when they are used for biting, or the mandible protrudes. This study is certainly only a snapshot of the processes that occur during mastication, and we are looking forward to seeing future studies including more mastication scenarios.

Text Change: Figure 6 (lines 198-203): Distribution of the forces under axial load (Pressure stress – positive values, tensile stress – negative values). Under simulated axial load, the pressure forces in the periodontium of the individual teeth are shown in the upper row diagrams (A). The upper first molars, and the second and first lower molars therby carry the most forces, followed by the premolars, and the canines (B). The anteriors are only minimally impacted by axial load. Teeth are numbered according to the Universal Numbering System.

3.4) Line 173: which teeth are reported and presented in Figure 7?

Response: Thank you for pointing out the imprecise or incomplete data presentation. We added more language and an additional figure (Fig. 2) to illustrate which teeth are reported in former Figure 7, now Figure 9.

Text Change: Figure 2 was added to show how the model for the reduced PDL analysis was created. This will hopefully also clarify the flow in the result presentation. Figures 2 and 7-9 show the analyzed teeth. 

(Lines 113-117) To investigate the behavior of a reduced PDL layer, 10 additional models of single- and multi-rooted teeth were created (Figure 2) of which each model was given a reduction of 1mm in PDL layer height along the root axis.

(Lines 121-124) Figure 2: CAD-formatted 3D model (A) including posterior single- and multi-rooted teeth for the reduced periodontium analysis. The FEA model (B) shows that the forces were applied from the teeth in the maxilla (1) towards the teeth and bone in the mandible (2).

(Lines 241-246) Figure 7 9: Graphical illustration of the correlations between increasing force and reduction of PDL on the pressure development for single (A) and multi-rooted (B) tooth (as shown in Fig. 7). Illustration depicts a single-rooted tooth (left hand) and amulti-rooted tooth (right hand) as shown in Figure 8. In general, pressure on PDL increases with an increasing force as well as with an increasing PDL reduction.

4) Comments on the Quality of English Language: Some improvements must be made to the clarity of the language and the technical details.

Response: We thank you for the critical and careful review of our manuscript. The manuscript was rechecked for errors by a native English speaker after the changes were made.

Text Change: English proof reading was performed. All changes are also highlighted in red.

Reviewer 3 Report

Comments and Suggestions for Authors

Dear Author

Congratulation for your efforts

It's a great work very interesting

Please can you improve MDPI references ? Can you change some one?

In the conclusion can you added what relevance or application may your discovery have in the dental clinic ?

Author Response

Manuscript ID bioengineering-2638967

Functional Load Capacity of Teeth with Reduced Periodontal Support: A Finite Element Analysis

Dear Professor Dr. Kwong Ming Tse,

We would like to thank Bioengineering for the careful evaluation of our manuscript, and we thank the reviewers for their valuable suggestions. The manuscript has been revised according to reviewers’ comments and the paper has much improved because of the changes that were suggested.

*Please note that changes have been highlighted in red in the revised manuscript.

Comments and Suggestions from Reviewer 3:

Dear Author

Congratulation for your efforts

It's a great work very interesting

1) Please can you improve MDPI references? Can you change some one?

Response: First, we would like to thank the reviewer for the time and effort in reviewing and critically evaluating of our manuscript. We are very pleased that the reviewer supports the relevance of our manuscript and are pleased with the positive feedback. Thank you for the suggestion to improve the literature selection. We have conducted a targeted literature search and added appropriate MDPI references.

Text Change: The following MDPI references were added to the manuscript:

  • Moga RA, Olteanu CD, Daniel BM, et al.: Finite elements analysis of tooth—A comparative analysis of multiple failure criteria. Int J Environ Res Public Health. 2023; 20: 4133
  • Sioustis I-A, Axinte M, Prelipceanu M, et al.: Finite Element Analysis of Mandibular Anterior Teeth with Healthy, but Reduced Periodontium. Appl Sci. 2021; 11: 3824.
  • Moga RA, Olteanu CD, Buru SM, et al.: Cortical and Trabecular Bone Stress Assessment during Periodontal Breakdown–A Comparative Finite Element Analysis of Multiple Failure Criteria. Medicina. 2023; 59: 1462.
  • Moga RA, Olteanu CD, Botez M, et al.: Assessment of the Maximum Amount of Orthodontic Force for Dental Pulp and Apical Neuro-Vascular Bundle in Intact and Reduced Periodontium on Bicuspids (Part II). Int J Environ Res Public Health. 2023; 20: 1179
  • Huang H-L, Tsai M-T, Yang S-G, et al.: Mandible integrity and material properties of the periodontal ligament during orthodontic tooth movement: a finite-element study. Appl Sci. 2020; 10: 2980.
  • Yoon Y, Lee M-J, Kang I, et al.: Evaluation of Biomechanical Stability of Teeth Tissue According to Crown Materials: A Three-Dimensional Finite Element Analysis. Materials. 2023; 16: 4756.
  • Richert R, Farges J, Tamimi F: Validated finite element models of premolars: A scoping review. Materials 13, 3280. 2020.

2) In the conclusion can you added what relevance or application may your discovery have in the dental clinic?

Response: Thank you for this suggestion. We have added the clinical relevance of the study to the conclusions and highlighted them.

Text Change:

Conclusions: Functional stress was highest under axial masticatory load in molars, followed by premolars and canines. In case of full periodontal support, the zones of highest stress were in the cervical and apical areas, whereas in the multi-rooted teeth, a significantly increased pressure load was also found in the furcation area. When the area of the PDL was successively reduced to simulate attachment loss, the pressure load on the remaining PDL increased approximately logarithmically and shifted significantly towards the apical region. A single-rooted tooth with attachment loss of >7 mm was predicted to fail, and a multi-rooted tooth of >8 mm attachment loss was predicted to fail. These findings suggest that there are thresholds of 7 mm in single-rooted teeth and 8 mm in multi-rooted teeth that can determine when a tooth is hopeless due to periodontal attachment loss, which can help the clinician in the decision making of when to extract. These thresholds can also be applied to assess the quality of a tooth as a potential abutment tooth for prosthetics.